# Practical Application of Urinary Zearalenone Monitoring System for Feed Hygiene Management of a Japanese Black Cattle Breeding Herd—The Relationship between Monthly Anti-Müllerian Hormone and Serum Amyloid A Concentrations

**DOI:** 10.3390/toxins14020143

**Published:** 2022-02-16

**Authors:** Oky Setyo Widodo, Makoto Etoh, Emiko Kokushi, Seiichi Uno, Osamu Yamato, Dhidhi Pambudi, Hiroaki Okawa, Masayasu Taniguchi, Mirni Lamid, Mitsuhiro Takagi

**Affiliations:** 1Joint Graduate School of Veterinary Sciences, Yamaguchi University, Yamaguchi 753-8515, Japan; oky.widodo@fkh.unair.ac.id (O.S.W.); masa0810@yamaguchi-u.ac.jp (M.T.); 2Department of Animal Husbandry, Faculty of Veterinary Medicine, Airlangga University, Surabaya 60115, Indonesia; mirnylamid@fkh.unair.ac.id; 3Ohita Agricultural Mutual Aid Association, Takeda 878-0024, Japan; mako_eto@nosai-oita.jp; 4Faculty of Fisheries, Kagoshima University, Kagoshima 890-0056, Japan; kokushi@fish.kagoshima-u.ac.jp (E.K.); uno@fish.kagoshima-u.ac.jp (S.U.); 5Joint Faculty of Veterinary Medicine, Kagoshima University, Kagoshima 890-0065, Japan; osam@vet.kagoshima-u.ac.jp; 6Department of Mathematics Education, Faculty of Teacher Training and Education, Sebelas Maret University, Surakarta 57126, Indonesia; dhidhipambudi@staff.uns.ac.id; 7Guardian Co., Ltd., Kagoshima 890-0033, Japan; okawa0117@guardian-vet.com; 8Laboratory of Theriogenology, Joint Faculty of Veterinary Medicine, Yamaguchi University, Yamaguchi 753-8515, Japan

**Keywords:** AMH, cattle, long-term monitoring, sub-clinical contamination, SAA, urine, zearalenone

## Abstract

This study addresses an advantageous application of a urinary zearalenone (ZEN) monitoring system not only for surveillance of ZEN exposure at the production site of breeding cows but also for follow-up monitoring after improvement of feeds provided to the herd. As biomarkers of effect, serum levels of the anti-Müllerian hormone (AMH) and serum amyloid A (SAA) concentrations were used. Based on the results of urinary ZEN measurement, two cows from one herd had urinary ZEN concentrations which were two orders of magnitude higher (ZEN: 1.34 mg/kg, sterigmatocystin (STC): 0.08 mg/kg in roughages) than the levels of all cows from three other herds (ZEN: not detected, STC: not detected in roughages). For the follow-up monitoring of the herd with positive ZEN and STC exposure, urine, blood, and roughage samples were collected from five cows monthly for one year. A monitoring series in the breeding cattle herd indicated that feed concentrations were not necessarily reflected in urinary concentrations; urinary monitoring assay by ELISA may be a simple and accurate method that reflects the exposure/absorption of ZEN. Additionally, although the ZEN exposure level appeared not to be critical compared with the Japanese ZEN limitation in dietary feeds, a negative regression trend between the ZEN and AMH concentrations was observed, indicating that only at extremely universal mycotoxin exposure levels, ZEN exposure may affect the number of antral follicles in cattle. A negative regression trend between the ZEN and SAA concentrations could also be demonstrated, possibly indicating the innate immune suppression caused by low-level chronic ZEN exposure. Finally, significant differences (*p* = 0.0487) in calving intervals between pre-ZEN monitoring (mean ± SEM: 439.0 ± 41.2) and post-ZEN monitoring (349.9 ± 6.9) periods were observed in the monitored five cows. These preliminary results indicate that the urinary ZEN monitoring system may be a useful practical tool not only for detecting contaminated herds under field conditions but also provides an initial look at the effects of long-term chronic ZEN/STC (or other co-existing mycotoxins) exposure on herd productivity and fertility.

## 1. Introduction

Recently, increasing attention has been paid to the impact of *Fusarium*-derived mycotoxins, as their prevalence seems to increase worldwide, despite efforts to minimize their concentration in animal feeds. Global warming and/or climate change are discussed as possible causes for higher exposure rates which may enhance the risk of harmful effects on both human and animal health [1,2,3,4]. Indeed, a large-scale global survey of mycotoxin contamination in more than 70,000 sample feeds collected from more than one hundred countries suggested that mycotoxins are almost ubiquitously detected contaminants [5]. The authors concluded that co-occurrence of *Fusarium*-derived mycotoxins (such as zearalenone (ZEN) and deoxynivalenol (DON) as the most important combinations) should be monitored more closely [5]. Based on a recent review, ZEN, one of the *Fusarium*-derived estrogenic-mycotoxins, is mainly formed at the pre-harvest stage, although continued fungal growth and ZEN synthesis may continue during poor storage conditions [6]. As controlling animals’ exposure to mycotoxins is often difficult under farm conditions, where feed supplies may change rapidly, monitoring of urinary concentration of ZEN of farm animals such as cattle is presumed to be a suitable biomarker for ZEN exposure [7,8,9]. Previously, we have established a urinary ZEN-monitoring system with ELISA for initial screening purposes, followed by LC-MS/MS validation to detect ZEN and its related metabolites; α-zearalenol (α-ZEL) and β-zearalenol (β-ZEL) as well as sterigmatocystin (STC), as both toxins might occur together in diets for Japanese cattle [10,11,12]. Additionally, we have reported that monitoring ZEN or STC levels in urine is not only a practical and useful way of evaluating and detecting the naturally contamination status of cattle herds, but also assessing the efficiency of mycotoxin adsorbents (MAs) supplemented in dietary feed to reduce intestinal absorption of mycotoxins [10,11,13,14].

Multiple factors influence fungal growth and mycotoxin formation, including season, geographical location, drought, harvest time, processing, storage, and distribution, etc., [6], thus, the important first step in combating mycotoxins, especially in herds fed home-grown forage begins with measuring the level/status of mycotoxins contamination in the feed of individual herds during the stage of sub-clinical health condition. We have continued to monitor cattle herds in the field to detect subclinical ZEN-contaminated herds by using urinary ZEN monitoring. During this monitoring, we identified one herd which was speculated to have been fed rather high ZEN-contaminated roughage exceeding the standard value in Japan (>1 mg/kg) with urine samples by ELISA, following validated by LC-MS/MS assay of the dietary roughage. Given ethical and animal welfare concerns, and the high costs involved, it is hardly possible to conduct feeding trials with cattle exposed to ZEN contaminated feed to investigate the effects of chronic low levels of ZEN contamination. Therefore, we evaluated whether the identified cattle herd may serve as a useful tool for observing the effect of long-term exposure on urinary excretion of toxins as well as an indicator of the reproductive performance of female cattle.

The objectives of this field study were to (1) re-evaluate the urinary ZEN monitoring system for its practical usefulness in cattle farm conditions and (2) evaluate the follow-up monitoring results for 1 year, concomitant with the relationship between changes in both naturally occurring urinary ZEN and serum anti-Müllerian hormone (AMH) concentration, and STC concentrations. AMH is secreted by ovarian granulosa cells primarily from pre-antral and early antral follicles of females and is an endocrine marker closely associated with both gonadotrophin-responsive ovarian reserves and with the size of the pool of growing preantral and small antral follicles [15,16]. Additionally, serum amyloid A (SAA), which is one of the most reliable acute phase proteins (APPs) primarily produced by the liver induced by the inflammatory cytokines, such as interleukin (IL)-1, IL-6, and tumor necrosis factor (TNF)-α [17,18], was measured to monitor inflammation in each cow at the monthly sampling time, given our previous report that not only calving itself but also severe inflammation during the postpartum period indicated by high SAA concentration can affect the AMH concentration in cows [19].

## 2. Results

### 2.1. First Urinary ZEN Screening on Four JB Breeding Cattle Herds in the Neighborhood

Table 1 shows a summary of all results of the first screening of the four herds. The urinary ZEN concentrations of two samples from Herd C measured by ELISA exceeded the upper limit value that guarantees quantification within the range of the calibration curve used in the ELISA measurement (4050 ppt = 4050 pg/mL); thus, they were assumed to be >20,250 pg/mL without repeating ELISA measurements by using more diluted samples, whose concentration levels differed by two orders of magnitude compared to cows from the other three herds. Therefore, based on the results of the first urinary ZEN screening, it was assumed that the ZEN concentration in the roughage fed to herd C was much higher than that in the other herds. Thus, as the next step in our screening, both ZEN and STC concentrations in the dietary roughage from all herds were measured. The ZEN concentration of the roughage sample from Herd C was 1.34 mg/kg, which was higher than the Japanese national limit of ZEN, concomitant with STC exposure (0.08 mg/kg) (Table 1). Additionally, the results of LC/MS measurements performed later, i.e., the simultaneous detection of ZEN, its metabolites, and STC in the urine sample only from Herd C, clarified the ELISA results of urine samples and LC-MS/MS results of both ZEN and STC in roughage.

### 2.2. Follow-up Monthly Monitoring in ZEN Detected JB Breeding Cattle Herd

Sampling could not be performed for Cow 4 in August 2020 because she was about to calve at the time of sampling. The monthly changes in both urinary ZEN concentrations measured by ELISA and serum AMH concentration are shown in Figure 1. During the 1-year follow-up period, two peaks of urinary ZEN concentrations were observed in August 2020 and between April and May 2021 with different concentrations in each cow, which were later confirmed by urinary ZEN and metabolite detection by LC-MS/MS measurement (Figure 1g). Additionally, STC was also detected in urine samples from four cows in July 2020 (Cow 1: 184.8), February (Cow 4: 508.5), March (Cow 1: 429.4, and Cow 2: 495.7), and one in June 2021 (Cow 1: 342.5) (Figure 1g). Alternatively, ZEN was only detected in May 2021 (0.03 mg/kg), and STC was detected in July 2020 (0.01 mg/kg), February (0.02 mg/kg), April (0.03 mg/kg), and May 2021 (0.05 mg/kg), suggesting that feed ZEN exposure does not correspond with urinary ZEN concentrations, and feed STC exposure does not seem to correspond with urinary STC concentrations (Figure 1h), which must be due to largely reflected by the influence of the sampling parts collected as roughage samples. 

The estimated values of ZEN, AMH, and SAA for each month estimated by linear mixed model analysis are shown in Table 2 and Figure 1f. The ZEN value peaked in August, then decreased from September to March, and trended upward from April. Conversely, the AMH value dropped once in September, trended upward in March, then trended downward again in April. The SAA also showed a trend of increasing until March of the following year, although there have been increases and decreases since August. 

The results of the time-series regression between the ZEN and AMH values are shown in Table 3. Although neither correlation was significant, the effect of the ZEN value one month earlier on AMH displayed a negative regression trend (β = −0.449 [−1.112, 0.214], *p* = 0.160 in lag 1 month model). In other words, a low ZEN value one month prior tended to result in a high AMH value in the current month. The results of the examination of the time-series regression between the ZEN and AMH change values are shown in Table 4. Although no correlations were significant, a negative correlation trend was observed for the effect of the ZEN value from one month before AMH change (β = −0.377, lag 1 month). In other words, a low ZEN value in the current month suggested a tendency for AMH values to be higher in the next month. 

The results of the time series regression between SAA and ZEN values are shown in Table 5. Although both regressions were non-significant, the effect of the current month’s ZEN value on SAA showed a negative regression trend (β = −0.400 [−1.046, 0.246], *p* = 0.198 in lag 0 model). In other words, a high ZEN value may tend to result in a low SAA value in the current month. The results of the examination of the time series regression between SAA change and ZEN value are shown in Table 6. All regressions were non-significant, and the regression coefficients were small. 

The calving interval of the herd were 389.8 ± 35.3 (*n* = 10) in 2018, 471.7 ± 33.1 (*n* = 18) in 2019, 387.8 ± 15.0 (*n* = 19) in 2020, and 408.5 ± 24.6 (*n* = 20) in 2021, and tendency toward decreased calving intervals (*p* = 0.099) was observed between 2019 (pre-ZEN monitoring period) and 2020 (post-ZEN monitoring period). Table 7 shows the results of the calving intervals of the five cows examined during the pre- and post-ZEN monitoring periods. The number of calving intervals during the post-ZEN monitoring period (349.9 ± 6.9) was significantly lower (*p* = 0.0487) than the pre-ZEN monitoring period (439.0 ± 41.2).

## 3. Discussion

Currently, many reports have aimed to clarify and prevent the harmful effects of mycotoxins at each stage focused on three major factors. First, the characters, toxicity, and metabolites against the organs and/or systemic function of each mycotoxin (including emerging mycotoxins such as enniatin, beauvericine, and emodin) with both in vitro and in vivo approaches [20,21]. Second, the detection methods for these mycotoxins (including the case of multiple mycotoxins coexistence with different spp. of fungi) within dietary feeds and biological fluids, such as serum, urine, and milk, from animals [2,22,23,24]. Third, not only feed management and control strategies for fungal infection but also the degradation approaches by physical, chemical enzymatic, and biological methods to prevent the harmful effects of mycotoxins for the animals [1,3]. In cattle practice, acute exposure to high doses of mycotoxins is usually responsible for well-characterized clinical symptoms, such as reduced feed intake or diarrhea. Sub-chronic and chronic exposure to low doses has been less well characterized but is considered to be responsible for reduced performance, for reduced pathogen resistance, and more generally, for many of the causes of damaged health, and potentially the reproductive efficacy, of the herd [25]. Therefore, it is essential to first monitor the contamination status of mycotoxins in dietary feeds at each farm level to limit the exposure risks of mycotoxins. As previously suggested, one practical approach is to evaluate the feed contamination on each farm with an ELISA test kit for mycotoxin screening, followed by further validation of the suspected feed samples with LC-MS/MS [6]. Following these approaches, one objective of the present field trial/test was to 1) evaluate and apply the urinary ZEN monitoring system for its practical usefulness in cattle farm conditions. As expected, the results of our first screening indicated that (1) urinary ZEN measurements may be useful for monitoring or evaluating the level of intestinal absorption of ZEN from dietary feeds with follow-up by even small urinary samples (0.5 mL) from the same herd, (2) it was possible to detect ZEN naturally contaminated cattle herds by relatively high ZEN levels in feeds by ELISA as a rather simple method within the laboratory, concomitant with the coexistence of STC contamination of the dietary rice straw or WCS by following LC-MS/MS measurement, and (3) in a cattle herd (C) with confirmed ZEN exposure, monthly monitoring in the following year made it possible to monitor and control the exposure levels of dietary roughages derived from rice straw from within the same paddy field. To the best of our knowledge, this is the first practical verification test conducted in some cattle in which data are available encompassing the period from detection to follow-up using the urinary ZEN monitoring system.

The greatest advantage of using the urinary ZEN concentration monitoring system is that the ZEN concentration that is actually ingested and absorbed from the intestinal tract can be monitored and compared with other herds. As previously reported [12,26], the problem is that the concentration of mycotoxins produced in dietary feeds may vary greatly depending on the collection site of the feed sample to be collected. Indeed, in the present study, different results between the urinary ZEN concentration and the ZEN concentration in the roughages in August 2019 seem to clearly show this problem. Using the urinary ZEN monitoring system, it is possible to monitor the concentration of mycotoxins absorbed from the intestinal tract, and the absorbed concentration of mycotoxins may reflect the degree of contamination of the mycotoxins in the feed for each herd and its feed intake by animals. Since urinary ZEN concentration may be affected by the intake volumes of contaminated feeds, it seems to be a suitable method for monitoring and comparing mycotoxin exposure in cattle whose daily feed amount is fixed between each herd. In fact, in this study, although the urinary ZEN concentrations of the two heifers in Herd D (3862.4 and 1669.8) were like those of Herd A (3280.9 and 930.5), the urinary concentration was approximately 2 to 4 times higher than that of Herd B (983.7 and 979.8). Presumably, when comparing the daily feed volume, especially roughage, amounts for cows in Herd D were approximately half that of cows in Herd B. Naturally, the ZEN contamination level was lower than that of cows in Herd C, in which ZEN exposure was detected that time. However, in terms of the level of natural contamination of ZEN in roughage, that for cows in Herd D was higher than for cows in Herds A and B because the daily roughage feed for herd D was half that of Herds A and B. This demonstrates again that it can be inferred by performing ZEN monitoring to compare contamination levels within the rice straw in the present study. To investigate the effects of chronic mycotoxin exposure on the health status and productivity of livestock herds, a urinary mycotoxin monitoring system for monitoring mycotoxin intake from dietary feeds is indispensable. As we have demonstrated in this study, the simultaneous screening of cattle herds in the same area with similar breeding environments will be an important future strategy to understand the status of mycotoxin contamination of cattle herds. Additionally, the measurement results of ELISA and LC-MS/MS, the two measurement methods used for ZEN concentration measurement in this study, indicate that the urinary ZEN measurement using the ELISA method is an accurate, simple, and useful measurement method for evaluating the dynamics of mycotoxin infiltration at rather low concentrations and long-term chronic exposure. 

The second purpose of the present study was to evaluate both ZEN (Fusarium mycotoxin called pre-harvest mycotoxin) and STC (Aspergillus mycotoxin called post-harvest mycotoxin) dynamics in the dietary roughage (rice straw and WCS) using the naturally ZEN (also STC)- contaminated herd (Herd C) detected in August 2019 as a model/examined herd, mainly by urinary ZEN monitoring during the year from July 2020 to June 2021. In addition, ZEN and its metabolites have been suggested to cause apoptosis of granulosa cell/atresia of follicles in several animals [27,28,29,30], the relationship between urinary ZEN and AMH concentration during the monitoring period was studied to clarify the effects of ZEN exposure on AMH secretion from antral follicles. In this regard, similar to our recently reported decrease in AMH concentration during the peripartum period [19], all five examined JB cows displayed a clear decline in AMH concentrations in the month of their calving with a large range of SAA concentrations (Cow 1: 3.1 mg/L, Cow 2: 23.4 mg/L, Cow 3: 4.9 mg/L, and Cow 5: 2.9 mg/L; data not shown). Thus, we deleted all AMH and SAA concentration data for the calving month of each cow from our data set in the present study. As a result, although it became clear that there was a large variation in AMH concentration in each month among each individual cow during 1 year period in the blood samplings (Figure 1a–e), our results regarding the relationship between ZEN and AMH suggest that natural exposure level of ZEN may affect AMH concentrations, and thus, the AFC in cattle ovaries (Figure 1f, Table 3 and Table 4). Our results indicated that a low ZEN value one month prior may tend to result in a high AMH value in the current month, and a low ZEN value in the current month suggested a tendency for higher AMH values in the next month. Therefore, it was suggested that when AMH rises, it may be affected by the ZEN value of the previous month, and when AMH decreases, it may be affected by the ZEN value of the current month. Thus, the effects of ZEN on AMH secretion appeared early but recovery of AMH secretion after ZEN exposure may take some time. As an interesting result obtained from the present study, a negative regression trend between the concentrations of ZEN and SAA; a high ZEN value may tend to result in a low SAA value in the sampling month, were observed (Table 6 and Table 7). ZEN has been reported to have immunotoxicity in addition to its endocrine disrupting effects [31,32]. Previous reports indicated that ZEN exposure altered the hepatic cellular immune response, and suppressed the secretion of proinflammatory cytokines, such as IL-1, IL-6, and TNF-α [31,32,33]. Therefore, the negative regression trend between urinary ZEN and SAA concentrations obtained in the present study is possibly due to innate immune suppression of cows by low-level chronic ZEN exposure. In the future, it will be necessary to increase the number of cow herds monitored, expand the scope of monitoring, and clarify that improving the feed while detecting the naturally exposed herd will lead to an improvement in productivity. At the same time, field tests in the process of improving the mycotoxins level in naturally contaminated feed will be important indicators of animal health risks. 

Several incidences of STC contamination in food and feed (e.g., grains, grain-based products, maize, and rice) have also been reported in Japan [34,35,36,37]. Rice straw is considered one of the most important roughages used in the production of beef cattle in Japan, and STC is a major mycotoxin produced in rice. However, the harmful or chronic effects of STC on cattle are not well understood, and there are no regulations or control measures for this toxin in Japan. Previous measures of large-scale in-feed mycotoxins confirm a large difference in the types of mycotoxins when multiple mycotoxins were detected in feed coexist in each country and region of the country [5]. In the present study, ZEN and STC co-exposure in rice straw (WCS) was also confirmed in the area screened, another prefecture in the Kyushu area where we previously detected co-exposure to ZEN and STC. Our results elucidate the characteristics of mycotoxin co-contamination of rice straw produced in Japan and future research should further expand the scope of the survey to understand the characteristics and relationships between the two mycotoxins. 

ZEN and its metabolites exhibit distinct estrogenic properties that affect the reproductive system of several animal species, especially pigs [9,38,39]. In contrast, clinical signs of hyperestrogenism are not frequently observed in ruminating cows, and then only following the ingestion of highly contaminated silage or long-term exposure to contaminated feed materials [40,41,42]. In the present study, we simply compared the calving intervals of the monitored herd before and after introducing the urinary ZEN monitoring system and observed significantly reduced calving intervals of the herd. We previously reported the in vitro effects of acute ZEN exposure on bovine oocytes by using in vitro maturation, in vitro fertilization (IVF), and in vitro culture systems in cattle, and found that a high ZEN concentration (>1 mg/kg in the culture medium) might have a detrimental effect on the meiotic competence of bovine oocytes but does not affect fertilization and development after IVF [43]. Additionally, we reported that natural-feed ZEN contamination levels below the threshold value (i.e., below the maximum permissible ZEN concentration in Japan) did not affect embryo production in Japanese Black and Holstein cows undergoing superovulation [44]. Therefore, it was suggested that ZEN-contaminated feed affects the fertility of cattle by influencing the development of embryos in the uterus after implantation. In this study, the roughages harvested in 2019, which were fed prior to our first ZEN screening, were ZEN-contaminated, and the long-term use of the contaminated feed affected the calving interval. It is speculated that the introduction of the urinary ZEN monitoring system controlled ZEN contamination in the feeds, which shortened the calving interval of the herd. Obviously, further studies with an increased number of monitor herds in the field are needed.

In conclusion, our results demonstrate that the urinary ZEN monitoring system is an important practical tool, not only for detecting contaminated herds under field conditions but also for revealing the effects of long-term chronic ZEN/STC (or other co-existing mycotoxins) exposure on herd productivity and fertility. To date, several approaches have been developed to reduce mycotoxin contamination and exposure, including strategies involving agronomy, plant breeding and transgenics, biotechnology, toxin binding, and deactivating feed additives, and feed supplier/animal producer education [26]. As shown in the present field trial, herd management with a urinary ZEN monitoring system may be a possible novel concept for creating awareness among herd managers thereby preventing mycotoxin exposure in cattle herds. 

## 4. Materials and Methods

All experiments were conducted according to the guidelines and regulations for the protection of experimental animals and guidelines stipulated by Yamaguchi University, Japan (no. 40, 1995; approved on 27 March 2017) and informed consent was obtained from the farmers.

### 4.1. Chemicals and Solvents

ZEN was purchased from MP Biomedicals (Heidelberg, Germany). The metabolites α-ZEL and β-ZEL were purchased from Sigma (St. Louis, MO, USA). Stock solutions of ZEN, α-ZEL, and β-ZEL, each at a concentration of 1 μg/mL in methanol, were stored under light protection at 4 °C. STC was purchased from MP Biomedicals (Heidelberg, Germany). Stock solutions of 1 μg/mL STC in acetonitrile were stored in the dark at 4 °C, and high-performance liquid chromatography (HPLC)-grade methanol was purchased from FUJIFILM Wako Pure Chemical Co. (Osaka, Japan). β-Glucuronidase/arylsulfatase solution was purchased from Merck (Darmstadt, Germany). Sodium acetate was purchased from Kanto Chemical Co., Ltd. (Tokyo, Japan), and Tris was purchased from Nacalai Tesque Inc. (Kyoto, Japan). 

### 4.2. Screening by Urinary ZEN Monitoring to Detect Cattle Herds Fed with Dietary Roughage with Elevated ZEN Contamination

Before the rice harvest period in September 2019, this screening was conducted at the Japanese Black (JB) breeding cattle production site to monitor the extent of ZEN contamination of rice straw and/or whole crop silage (WCS) stored by cattle farmers in the summer season when the mean temperature of daytime is higher than 30 °C. At the request of the managing veterinarian, urinary ZEN monitoring was performed in four herds (A, B, C, and D) of JB cows kept for breeding in the neighborhood in the Kyushu area, Japan, for which the veterinarian routinely provides veterinary treatments and consults with four farmers. All animals were housed indoors, and roughage and concentrates were fed separately. Feeding and management systems were similar in each herd and the dates of sampling and contents of the feeds in each herd are detailed in Table 8. As feed intake may reflect the ZEN exposure, urine samples were collected from two cows with similar body weight within each herd during natural urination after softly massaging the perineum. Regarding the number of cows to be sampled for urine in each herd, referring to our previous report [10], we considered samples from two cows to be sufficient to evaluate and estimate the contamination status of feed fed the same amount and same lot of feed. In addition, samples of all roughages, such as rice straw and WCS, were obtained from each herd to measure both ZEN and STC concentrations in the roughage. All concentrates fed to cattle in each herd were purchased from feed companies and are generally tested for mycotoxin contamination during the manufacturing stage. The urine and roughage samples were immediately placed into a cooler, protected from light, transported to the clinic office, and frozen. The frozen samples were sent to our laboratory and stored at −30 °C until our analysis of ZEN and creatinine (Crea) concentrations in the urine, and ZEN and STC concentrations in the roughage.

### 4.3. Follow-up Monthly Monitoring on the Breeding Cattle Herd with Known Feed Contamination

Since contamination of rice straw/WCS from herd C collected in August 2019 exceeded the standard value of ZEN ≥ 1 mg/kg concomitant with STC was detected, this herd was selected for further monitoring. Therefore, monthly regular urinary ZEN monitoring of herd C was performed from July 2020 to help determine whether similar ZEN and STC exposure from the rice straw/WCS occurred year to year. For monitoring, five cows (Cows 1 to 5: mean 5.0 y: 3.6–6.3 y) in Herd C with similar body weight (approximately 500 kg) fed with the same roughage and concentrated feed were selected and monthly urine, blood, and roughage were sampled. We collected both urine and blood samples from the five cows at the beginning of each month, approximately 2 h after the morning feed, as per our previous methodology [10], and we also collected roughage samples fed to these cows. Both urine and blood samples were immediately stored on ice, protected from light, and transported to the laboratory, and were stored at –30 °C after centrifugation as dispensed urine and serum in microtubes until analysis. The collected roughages were also stored at –30 °C until measurement of both ZEN and STC concentrations. 

Zearalenone concentrations in the collected urine samples were measured by ELISA every two months, as described below, and urine samples were measured monthly when deemed necessary by the herd manager monitoring the condition of the roughage being fed or by contamination status of the roughage by fungi at the monthly sampling. During the follow-up period, daily feeding was performed while sharing the urinary ZEN concentration measurement results with the herd manager and the managing veterinarian. When a high urinary ZEN concentration was confirmed, the roughage lot fed at the time of sampling was changed, and the urinary ZEN concentration was measured again in the following month for follow-up purposes, concomitant with measurement of both ZEN and STC concentrations of roughage samples by LC-MS/MS as mentioned below. Concentrations of urinary ZEN, its metabolites, α-ZEL, β-ZEL, and STC of all collected urine samples during the follow-up period were measured by LC-MS/MS within one assay for reconfirmation of results by the ELISA assay and urinary STC measurement. A schematic representation of the experimental design is shown in Figure 2.

### 4.4. Reproductive Records

As a reproductive record, the calving intervals of the herd were compared for each year from 2017 to 2021. Additionally, the reproductive records from the five cows examined between pre-ZEN monitoring (2017 to 2019) and post-ZEN monitoring periods (2020 and 2021) were evaluated to confirm the impact of introducing the ZEN monitoring system on herd fertility.

### 4.5. Analytical Methods of ZEN in Urine and Feed Samples

Zearalenone concentration in urine was determined using a commercially available kit (RIDASCREEN Zearalenon; R-Biopharm AG, Garmstadt, Germany) according to the manufacturer’s instructions, with minor modifications. Briefly, a urine sample (0.1 mL: 5-fold dilution of the kit) was added into 3 mL of 50 mM sodium acetate buffer (pH 4.8) and the solution was incubated for 15 h at 37 °C in the presence of 10 μL of β-glucuronidase/arylsulfatase solution. Thereafter, the samples were loaded onto a C18 solid-phase extraction (SPE) column (Strata; Phenomenex, Torrance, CA, USA), which had been preconditioned with 3 mL of methanol, followed by 2 mL of 20 mM Tris buffer (pH 8.5)/methanol (80:20). After washing the SPE column with 2 mL of 20 mM Tris buffer (pH 8.5)/methanol (80:20) and 3 mL of methanol (40%), the column was centrifuged for 10 min at 500× *g* to dry the column. The analytes were then eluted slowly (flow rate: 15 drops/min) with 1 mL of methanol (80%). The eluate was evaporated to dryness at 60 °C using a centrifugation evaporator. The dried residue was redissolved in 50 μL of methanol, 450 μL of sample dilution buffer was added, the solution was mixed thoroughly, and an aliquot of 50 μL was used for the ELISA assay. To determine the ZEN concentration in the urine sample, RIDA SOFT Win (R-Biopharm) was used to calculate the absorbance at 450 nm using a microplate spectrophotometer. The cross-reactivity rates using this particular ELISA kit for α-ZEL, β-ZEL, and Zeranol were 41.6, 13.8%, and 27.7%, respectively, based on the manufacturer’s instruction, and the mean recovery rate of the ELISA assay based on the three trials was 84% ± 14%. 

Urine creatinine concentrations were determined using a commercial kit (Sikarikit-S CRE, Kanto Chemical, Tokyo, Japan), according to the manufacturer’s instructions, and were measured using a 7700 Clinical Analyzer (Hitachi High-Tech, Tokyo, Japan). All urine concentrations were expressed as a ratio of creatinine (pg/mg creatinine), as described previously [10].

Based on the results of the first screening and measurement of urinary ZEN concentrations by ELISA, both the urine and roughage samples in herds expected to have high ZEN infiltration in the feed were retested using a liquid chromatography-tandem mass spectrometry (LC-MS/MS) measurement system not only for the confirmation of the ELISA results but also for measuring the ZEN metabolites, α-ZEL and β-ZEL. Additionally, as per our previous reports, urinary STC levels were concomitantly higher in cattle fed ZEN-contaminated rice straw in Japan; thus, it was speculated that co-contamination of both ZEN and STC was observed. Therefore, in the retest, the STC concentration in urine and roughage was also measured according to our previous reports [10,12]. 

The LC-MS/MS method and validation have been described in our previous report [10]. Briefly, each urine sample (0.5 mL) was mixed with 3.0 mL of 50 mM ammonium acetate buffer (pH 4.8) and 8 μL of glucuronidase/arylsulfatase solution and incubated for 12 h at 37 °C. The solution was loaded onto a C18 SPE column, which was preconditioned with 3 mL 100% methanol and 2 mL Tris buffer, followed by the addition of 2 mL Tris buffer and 3 mL of 40% methanol. After washing the SPE column with approximately 1 mL of 80% methanol, the volume of the eluted solution was adjusted to 1 mL. Then, 20 μL of the reconstituted solution was injected into the LC-MS/MS system. The LC-MS/MS analyses were performed on an API 2000 MS/MS system (Applied Biosystems, Foster City, CA, USA) equipped with an electrospray ionization (ESI) interface and a 1200 Infinity Series HPLC system (Agilent Technologies, Santa Clara, CA, USA). The detection limits for ZEN, α-ZEL, and β-ZEL were 0.04 ng/mL, 0.05 ng/mL, and 0.05 ng/mL, respectively, while the mean recovery rates for ZEN, α-ZEL, and β-ZEL were 90%, 109%, and 90%, respectively. STC concentrations of the same eluted solution described above were also determined by LC-MS/MS using an API 2000 system equipped with an ESI as previously described [12]. Briefly, after elution with approximately 1 mL of 80% methanol, the volume was adjusted to exactly 1 mL, and 20 μL of the solution was injected into the LC-MS/MS system. Chromatographic separation was performed on an Inertsil ODS-3 column (4.6 i.d. × 100 mm, 5 μm; GL Sciences, Tokyo, Japan) at 40 °C. A mobile phase consisting of methanol/water/acetic acid (97:3:0.01, v:v:v) was used (200 μL/min) to separate the analyte in isocratic mode. Measurements were performed for 15 min. The limit of detection (LoD) was 0.2 ng/mL. ZEN, α-ZEL, β-ZEL, and STC concentrations in the urine are expressed as a ratio to creatinine (pg/mg creatinine). 

Both STC and ZEN concentrations in the roughage samples were measured using an API 3200 LC-MS/MS system (AB Sciex, Tokyo, Japan) equipped with an electrospray ionization (ESI) interface and a Prominence HPLC system (Shimadzu Corp., Kyoto, Japan), according to the Food and Agricultural Materials Inspection Center [45] at Shokukanken Inc., Gunma, Japan. In brief, representative samples of stored straw (2 g) and concentrate (10 g) were homogenized and chopped into small pieces. Each sample was placed in a sample tube, to which 20 mL of 84% acetonitrile was added. The tubes were shaken for 1 h and centrifuged for 10 min at 500× *g* at room temperature. The supernatant (10 mL) was loaded onto a MultiSep 226 Aflazon + multifunctional column (Romer Labs, Union, MO, USA). Subsequently, 1 mL of the eluent was mixed with 1 mL acetic acid (1 + 100) and centrifuged for 5 min at 500× *g*. Next, 10 μL of supernatant was injected into the LC-MS/MS system under the following conditions: column, Synergi 4 μm Polar-RP 80 A (2 mm × 150 mm, 4 μm); oven temperature, 40 °C; eluent flow, 200 μL/min; and solvent, methanol (A) + 1 mM Ammonium acetate in 0.1% aqueous acetic acid (B). An ESI probe was used in the positive mode for the STC analysis and the negative mode for the ZEN analysis. The detection limit for each analyte was 0.01 mg/kg. The mean STC and ZEN recovery rates were 90.5%–93.5% and 95.3%–98.5%.

### 4.6. Analytical Methods of AMH and SAA in Serum Samples

Serum AMH concentration was measured using a bovine AMH ELISA kit (AnshLabs, Webster, TX, USA), according to a previous report [46] to monitor the ovarian AFC of the examined cows during the follow-up period. Briefly, undiluted plasma (50 μL) was used for the assay, which had a limited detection of 11 pg/mL and a coefficient of variation of 2.9%, according to the manufacturer’s instructions. Based on our previous studies [19], it is clear that the blood AMH concentration in cattle is lower than usual during the peripartum period; thus, in this study, the AMH concentration in each cow’s calving month during the monitoring period was evaluated with particular care. Additionally, SAA concentrations were measured using an automated biochemical analyzer (Pentra C200; HORIBA ABX SAS, Montpellier, France) with a special SAA reagent for animal serum or plasma (VET-SAA ‘Eiken’ reagent; Eiken Chemical Co. Ltd., Tokyo, Japan) to monitor the inflammation status of each cow during sampling. The SAA concentration was calculated using a standard curve generated using a calibrator (VET-SAA calibrator set; Eiken Chemical Co. Ltd., Tokyo, Japan).

### 4.7. Data Management and Statistical Analysis

Monthly estimates for ZEN, AMH, and SAA were calculated using mixed model analysis with subject as a variable factor, because they contain missing data due to calving of the examined cows. Because the ZEN and SAA values approximate a lognormal distribution, the geometric mean estimate was calculated. The AMH value approximates a normal distribution; therefore, the arithmetic mean estimate was calculated. The effects of ZEN and AMH values were evaluated by calculating the simple regression of ZEN values with AMH and the time-lagged regression, which examines the effect of ZEN values one month earlier (lag 1 month), using a linear mixed model. Furthermore, the effects of ZEN on AMH change were evaluated by defining the change in AMH value over one month as the change from the previous month. The analysis was similarly evaluated by calculating the simple regression of ZEN value to AMH change (lag 0 model) and the time-lag regression to examine the effect of ZEN value one month earlier using a linear mixed model. In other words, the lag 1-month model evaluates the effect of the ZEN value of the current month on the AMH change until the next month. In addition, the effect of ZEN on SAA was also evaluated using the same linear mixed model as described above. A two-sided *p*-value ≤ 0.05 was considered statistically significant. All statistical analyses were performed using SPSS for Windows (version 24.0; IBM Japan, Tokyo, Japan).

All results of the reproductive records of the herds obtained are expressed as the mean ± SEM. Statistical analyses were performed using BellCurve for Excel software (Social Survey Research Information Co., Ltd., Tokyo, Japan). Calving intervals of the herd from 2017 to 2021 were compared using a one-way analysis of variance, followed by a post-hoc test (Tukey-Kramer). Additionally, calving intervals during the pre- (2017 and 2019) and post-ZEN monitoring (2020 and 2021) periods of the examined five cows were compared between the groups using Student’s t-test to determine the effects of introducing the monthly urinary ZEN monitoring system on the reproductive efficacies of the breeding herd. Statistical significance was set at *p* ≤ 0.05, whereas *p*-values ranging between 0.05 and 0.1 were considered to indicate a trend toward significance. 

## Figures and Tables

**Figure 1 toxins-14-00143-f001:**
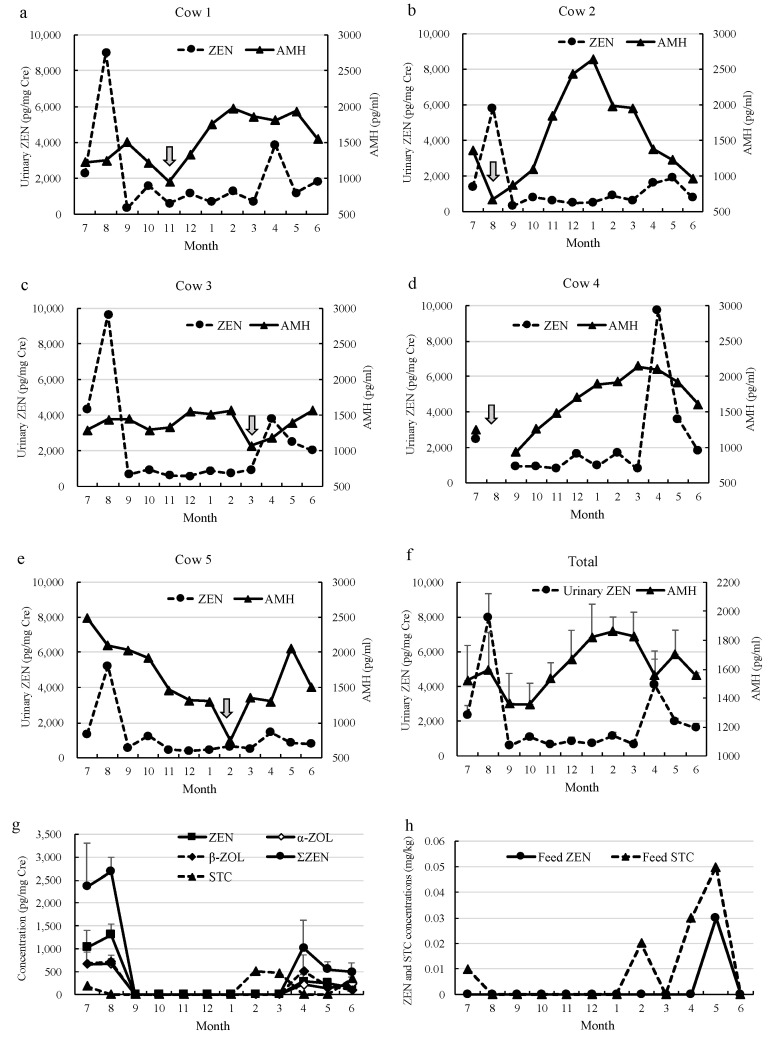
Monthly changes of both urinary ZEN concentration measured by ELISA and serum AMH concentration of each cow; (**a**) Cow 1, (**b**) Cow 2, (**c**) Cow 3, (**d**) Cow 4, and (**e**) Cow 5, 
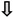
: calving, (**f**) total; mean urinary ZEN concentration and AMH from five cows, (**g**) monthly changes of urinary ZEN, its metabolites, and STC concentrations measured by LC-MS/MS, (**h**) monthly changes of ZEN and STC concentrations in the dietary roughage measured by LC-MS/MS.

**Figure 2 toxins-14-00143-f002:**
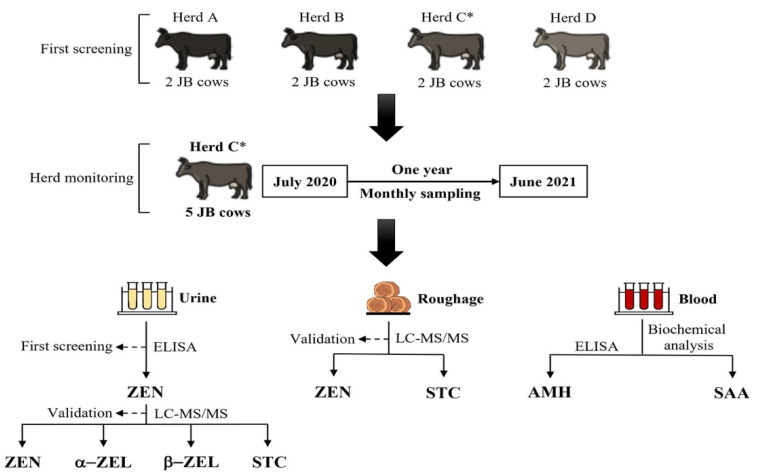
Schematic representation of the experimental design. C*, herd with high contamination: urinary ZEN concentrations exceeding the standard concentration of the ELISA kit, and ZEN-contaminated roughage exceeding the standard value in Japan (>1 mg/kg). α-ZEL: α-zearalenol; β-ZEL: β-zearalenol; AMH: anti-Müllerian hormone; SAA: serum amyloid A; STC: sterigmatocystin; ZEN: zearalenone.

**Table 1 toxins-14-00143-t001:** Urinary ZEN concentrations at the first screening on four JB breeding cattle herds.

	ELISA		LC-MS/MS					LC-MS/MS	
Cow	Urinary ZEN Concentrations (pg/mL)	ZEN/Cre	ZEN/Cre	α-ZEL/Cre	β-ZEL/Cre	ΣZEN/Cre	STC/Cre	ZEN in Roughage (mg/kg)	STC in Roughage (mg/kg)
A1	2132.6	3280.9	ND	ND	ND	ND	ND	ND	ND
A2	1637.6	930.5	ND	ND	ND	ND	ND		
B1	1937.9	983.7	ND	ND	ND	ND	ND	ND	<0.04 **
B2	1528.5	979.8	ND	ND	ND	ND	ND		
C1	>20,250	>23,011.4 *	14,363.6	10,772.7	16,454.5	41,590.9	659.1	1.34	0.08
C2	>20,250	>21,315.8 *	11,915.8	8526.3	4736.8	25,178.9	442.1		
D1	1931.2	3862.4	ND	ND	ND	ND	ND	ND	ND
D2	985.2	1669.8	ND	ND	ND	ND	ND		

* The urinary ZEN concentrations of the two samples from Herd C ranged over the maximal standard concentration of the ELISA kit. Thus, ZEN/Cre were expressed based on the maximal standard concentrations. Cre: Creatinine. ** Sterigmatocystin was detected below the lower limit, but it was not reached the quantitative value. ND: Not detected.

**Table 2 toxins-14-00143-t002:** Estimated means and confidence intervals of ZEN, AMH, and SAA at each time point by mixture model.

	ZEN		AMH		SAA	
Date	Geometric Mean	95% CI	Arithmetic Mean	95% CI	Geometric Mean	95% CI
2020/7	2142.2	1373.8–3340.4	1521.6	1165.7–1877.5	2.8	1.5–4.9
2020/8	8056.5	4853.3–13373.8	1594.5	1135.0–2053.9	2.5	1.2–5.2
2020/9	521.1	334.2–812.6	1358.0	1002.1–1713.9	2.9	1.6–5.1
2020/10	1065.7	683.4–1661.7	1356.6	1000.7–1712.5	4.0	2.2–7.1
2020/11	627.1	392.9–1000.9	1532.9	1134.9–1930.8	3.4	1.8–6.4
2020/12	720.8	462.2–1123.9	1665.8	1309.9–2021.7	3.1	1.7–5.4
2021/1	676.9	434.1–1055.4	1820.8	1464.9–2176.7	4.9	2.8–8.7
2021/2	995.8	623.9–1589.5	1860.0	1462.1–2257.9	3.1	1.6–5.8
2021/3	669.9	419.7–1069.2	1828.1	1430.1–2226.0	5.2	2.8–9.8
2021/4	3194.8	2048.8–4981.8	1553.8	1197.9–1909.7	3.3	1.9–5.9
2021/5	1763.7	1131.0–2750.2	1704.4	1348.5–2060.3	3.0	1.7–5.3
2021/6	1414.2	885.9–2257.7	1554.7	1156.8–1952.6	2.2	1.2–4.2

95% CI: 95% confidence interval.

**Table 3 toxins-14-00143-t003:** Regression between ZEN and AMH values.

	AMH				
	β	95% CI			*p*-Value
Simple correlation					
ZEN	−0.085	−0.787	–	0.617	0.793
Time-lagged correlation					
ZEN (lag 1 month)	−0.449	−1.112	–	0.214	0.160

The effects of ZEN on AMH values were evaluated by calculating the simple regression of ZEN values to AMH and the time-lagged regression, which examines the effect of ZEN values, one month earlier (lag 1 month), using a linear mixed model. β: Standardized regression coefficient. 95% CI: 95% confidence interval.

**Table 4 toxins-14-00143-t004:** Regression between ZEN and AMH changes.

	AMH Change over one Month				
	β	95% CI			*p*-Value
Time-lagged correlation					
ZEN (lag 0)	−0.024	−0.744	–	0.695	0.941
ZEN (lag 1 month)	−0.377	−1.039	–	0.285	0.230

The analysis was similarly for Table 3 evaluated by calculating the simple regression of ZEN value to AMH change over one month (lag 0 model) and time-lag regression to examine the effect of ZEN value one month earlier using a linear mixed model.

**Table 5 toxins-14-00143-t005:** Regression between ZEN and SAA values.

	SAA				
	β	95% CI			*p*-Value
Simple regression					
ZEN	−0.400	−1.046	–	0.246	0.198
Time-lagged regression					
ZEN (lag 1 month)	−0.333	−1.029	–	0.364	0.308

The effects of ZEN on SAA values were evaluated by calculating the simple regression of ZEN values to SAA and the time-lagged regression, which examines the effect of ZEN values one month earlier (lag 1 month), using a linear mixed model. β: standardized regression coefficient. 95%CI: 95% confidence interval.

**Table 6 toxins-14-00143-t006:** Regression between ZEN and SAA changes.

	SAA Change over one Month				
	β	95% CI			*p*-Value
Time-lagged regress					
ZEN (lag 0)	−0.245	−0.941	–	0.450	0.446
ZEN (lag 1 month)	0.065	−0.654	–	0.784	0.843

The analysis was similar for Table 5 evaluated by calculating the simple regression of ZEN value to SAA change over one month (lag 0 model) and the time-lag regression to examine the effect of ZEN value one month earlier using a linear mixed model.

**Table 7 toxins-14-00143-t007:** Mean calving intervals of the examined 5 cows during pre- and post-ZEN monitoring periods.

	Birthday	2017 (Pre)	2018 (Pre)	2019 (Pre) *	2020 (Post) **	2021 (Post)
Cow 1	9 January 2016	-	351	335	349	333
Cow 2	8 November 2014	690	-	380	321	349
Cow 3	7 April 2014	346	392	437	334	346
Cow 4	15 July 2015	-	600	-	377	355
Cow 5	27 December 2016	-	-	420	-	385
Mean of 5 cows		518.0 ± 172.0	447.7 ± 77.1	393.0 ± 22.7	345.3 ± 12.0	353.6 ± 8.6
Mean of the pre- and post- monitorin		439.0 ± 41.2 ^a^ (*n* = 9)			349.9 ± 6.9 ^b^ (*n* = 9)	

* Pre: Pre-ZEN monitoring period, ** Post: Post-ZEN monitoring period. ^a,b^: *p* < 0.05.

**Table 8 toxins-14-00143-t008:** Composition of feeds provided to the monitored herds kept for breeding purposes.

Herd	Date of Sample Collection	Forage Feeds/Day	Formula Feeds/Day
A (*n* = 2) (Both 12 y) *	10 July 2019	Home-grown rice straw 2 kg, Home-grown WCS (rice) 6 kg, Home-grown Italian ryegrass 4 kg Total: 12 kg	Commercially available concentrates 4 kg
B (*n* = 2) (3 y and 5 y)	24 June 2019	Home-grown rice straw 10 kg, Mixed of Italian ryegrass and Orchard grass 10 kg Total: 20 kg	Commercially available concentrates 1 kg, Wheat bran 1 kg, Maize 1 kg
C (*n* = 2) (8 y and 10 y)	19 August 2019	Home-grown rice straw 12~14 kg, Orchard grass 10 kg (once a week) Total: 12~14 kg	Commercially available concentrates 3 kg Wheat 0.5–1 kg
D (*n* = 2) (9 m and 10 m)	11 July 2019	Imported Oats-hey 2.25 kg, Bermuda-grass 2.25 kg Total: 4.5 kg	Commercially available concentrates 4.5 kg

* Age of the breeding cattle at sampling, y; years old, m; month old. WCS: whole crop silage.

## Data Availability

The original contributions presented in the study are included in the article/supplementary material; further inquiries can be directed to the corresponding author.

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
