# Peer review of "Practical Application of Urinary Zearalenone Monitoring System for Feed Hygiene Management of a Japanese Black Cattle Breeding Herd—The Relationship between Monthly Anti-Müllerian Hormone and Serum Amyloid A Concentrations"

_toxins, 2022, doi:10.3390/toxins14020143_

Round 1
Reviewer 1 Report
Title: Practical Application of Urinary Zearalenone Monitoring System for Feed Hygiene Management of a Japanese Black Cattle Breeding Herd
Interesting study on the monitoring of zearalenone in the urine of the Japanese Black cattle. I very much appreciate that this is a long-term monitoring of this metabolite, which is still one of the most important toxins causing significant economic losses in cattle. In addition to the small comments that are listed below this, I have one complain for the number of cows per group, when two pieces do not seem adequate to me. Statistically, such data are basically invaluable, because, for example, it is clear from Table 2 that each individual has a different value, which sometimes differs multiple times. However, if you only want to prove that these were positive samples, then I understand that the number of two pieces is sufficient for you. Above all that, I consider this study to be highly important and beneficial, so I recommend this manuscript for publication with some minor changes as listed below.
Specific comments:
Keywords: repeating the title, should be reconsidered
37: „Li et al., 2020b“ – should be and “a” due to it’s the first place where the “Li et al., 2020” is referred
38-39: missing reference
94: delete the colon “:” at the end of title
309: bounced line
324: missing space
326: “May” with small letter; missing space
Figure 2: missing the labelling of tables (g,h); the explanations of “ZEN” and “AMH” should be given in all of the figures a-h (is missing for b to f)
331-341: Why the ZEN concentrations detected in feed do not correspond to the urinary ZEN concentrations? Why the elevations are not in the similar months?
Table 3: why the arithmetic mean is used in AMH evaluation and the geometric mean for ZEN and SAA?
410: correct the reference
Author Response
Toxins
Manuscript ID: toxins-1583021
Title:
Practical Application of Urinary Zearalenone Monitoring System for Feed Hygiene Management of a Japanese Black Cattle Breeding Herd
– the Relationship Between Monthly Anti-Müllerian Hormone and Serum Amyloid A Concentrations
We have revised the manuscript in accordance with the suggestions of Reviewer 1 as follows. Both the revised sections of the manuscript and the responses to the Reviewer below are marked in red.
Specific Comments
Keywords: repeating the title, should be reconsidered
Thank you very much for your insightful comment. Accordingly, we revised the Keywords as follows; AMH; cattle; long-term monitoring; sub-clinical contamination; urine; zearalenone. Line 29.
37: „Li et al., 2020b“ – should be and “a” due to it’s the first place where the “Li et al., 2020” is referred
Thank you for this observation. Accordingly, we corrected this issue and reordered the References, as needed. Lines 37, 428, 494, 618-621.
38-39: missing reference
Thank you for this observation. The missing reference was added, accordingly. Lines 39-40.
94: delete the colon “:” at the end of title
Per your recommendation, we deleted the colon. Line 94.
309: bounced line
Thank you for this remark. This issue has addressed accordingly. Lines 307-310.
324: missing space
Thank you for this remark. Accordingly, we added the missing space. Line 324.
326: “May” with small letter; missing space
We apologize for this oversight. We have addressed the capitalization issue and inserted a space. Line 326.
Figure 2: missing the labelling of tables (g,h); the explanations of “ZEN” and “AMH” should be given in all of the figures a-h (is missing for b to f)
Thank you very much for your observation. We have added both ZEN and AMH to the legends for Figure 2 b to f.
331-341: Why the ZEN concentrations detected in feed do not correspond to the urinary ZEN concentrations? Why the elevations are not in the similar months?
Thank you very much for your insightful comments. Previous reports indicated that mycotoxins are not evenly distributed in the feed matrix, and feed intake is difficult to estimate accurately (Baranski et al., Toxins 2021). Therefore, we assume that the concentration of mycotoxins in the feed reflects the amount of mycotoxins present at the site where the feed was sampled, which is considered to be the cause of the difference in concentration of mycotoxins in the urine samples.
Table 3: why the arithmetic mean is used in AMH evaluation and the geometric mean for ZEN and SAA?
Thank you for this remark. As we mentioned in the Materials and Methods section, because the ZEN and SAA values approximate a lognormal distribution, the geometric mean estimate was calculated. However, the AMH value approximates a normal distribution, therefore, the arithmetic mean estimate was calculated. We appreciate your kind understanding.
410: correct the reference
We apologize for this oversight. We have revised it (deleted “L”) as is appropriate. Line 424.

Reviewer 2 Report
Did you measure ZEN values in the feed as well?
How do you then explain that there are higher ZEN values in one herd?
Although the analytical methods were chosen appropriately, the number of individuals examined per herd is insufficiently representative and cannot be statistically evaluated. The number of individuals does not correspond to the smallest relevant statistically evaluable group of individuals.
Author Response
Toxins
Manuscript ID: toxins-1583021
Title:
Practical Application of Urinary Zearalenone Monitoring System for Feed Hygiene Management of a Japanese Black Cattle Breeding Herd
– the Relationship Between Monthly Anti-Müllerian Hormone and Serum Amyloid A Concentrations
We have revised the manuscript in accordance with the suggestions of Reviewer 2 as follows. The responses to the Reviewer below are marked in blue.
Specific Comments
Did you measure ZEN values in the feed as well?
Thank you for this insightful remark. As we mentioned in the text, all concentrates (except for roughage) fed to cattle in each herd were purchased from feed companies and are generally tested for mycotoxin contamination, including ZEN, during the manufacturing stage (Lines 124-125). Additionally, the consumption period of each purchased concentrate on each farm is rather short. Therefore, we have determined that the possibility of ZEN contamination of the concentrates is extremely low (indirectly demonstrated by the reduction of urinary ZEN concentration by changing the contaminated roughage to non-contaminated in our monitoring). Thus, we did not measure the mycotoxin concentration of concentrates in this study. Additionally, this also means that by making full use of the information on feed obtained from farmers, in the field, this monitoring system will be simple and inexpensive.
How do you then explain that there are higher ZEN values in one herd?
Thank you for this insightful remark. As described in the manuscript, our analyses, including the screening results of 4 cow herds in the present study, regarding the urine ZEN measurement results in the herd that we have conducted so far indicate that when ELISA measurement is performed using 5-fold diluted urine as a sample, ZEN infiltration in the feed exceeding the regulation value is suspected when the concentration exceeding the measurement limit value of the kit is detected, following the validation of the dietary feed inspection with LC-MS/MS. Therefore, it is also applicable to monitoring one herd, and in fact, we are conducting such an application in the field to detect ZEN contaminated herds in dietary feed (roughage).
Although the analytical methods were chosen appropriately, the number of individuals examined per herd is insufficiently representative and cannot be statistically evaluated. The number of individuals does not correspond to the smallest relevant statistically evaluable group of individuals.
Thank you very much for this insightful remark. In fact, Referee 1 also pointed out a similar shortage of statistical samples. The most important purpose of the first monitoring performed with the four herds in the present study was to use urine samples to detect cattle herds with a strong suspicion of ZEN contamination in the feed, as described above, simply and inexpensively under farming conditions. Finally, we could indicate that, although the number of individuals examined per herd was insufficient for statistical evaluation, at least 2 urine samples from the cattle herd under the same feeding condition may be sufficient to provide the ZEN positive samples, following validation for the dietary feed examinations. Therefore, although we cannot perform statistical processing due to the small number of samples based on the first screening results, we are confident that we can obtain information on the ZEN contamination status, which is more important in bovine clinical practice. We appreciate your kind understanding.

Round 2
Reviewer 2 Report
An explanation is sufficient.
In the discussion, it would be worth mentioning the effective additive against fusarium mycotoxins in feeds (DOI: 10.1007 / s11356-020-10581-x).
Author Response
Toxins
Manuscript ID: toxins-1583021
Title:
Practical Application of Urinary Zearalenone Monitoring System for Feed Hygiene Management of a Japanese Black Cattle Breeding Herd
– the Relationship Between Monthly Anti-Müllerian Hormone and Serum Amyloid A Concentrations
We have revised the manuscript in accordance with the suggestions of Reviewer 2 as follows. The responses to the Reviewer below are marked in blue.
Specific Comments
In the discussion, it would be worth mentioning the effective additive against fusarium mycotoxins in feeds (DOI: 10.1007 / s11356-020-10581-x).
Thank you for this insightful remark. In our manuscript, we already mentioned the use of additives as a mycotoxin control method not only in Introduction and Discussion section, but also in conclusion part as control methods currently considered on farm conditions. Therefore, we would like to maintain the current contents of our manuscript. We appreciate your kind understanding.
